# Tuning Legged Locomotion Controllers via Safe Bayesian Optimization

**Daniel Widmer**[*]        **Dongho Kang**[*]        **Bhavya Sukhija**

**Jonas Hübotter**        **Andreas Krause**        **Stelian Coros**

ETH Zürich
`{widmdani, kangd, sukhijab, jhuebotter, krausea, scoros}@ethz.ch`

**Abstract:** This paper presents a data-driven strategy to streamline the deployment of model-based controllers in legged robotic hardware platforms. Our approach leverages a model-free safe learning algorithm to automate the tuning of control gains, addressing the mismatch between the simplified model used in the control formulation and the real system. This method substantially mitigates the risk of hazardous interactions with the robot by sample-efficiently optimizing parameters within a probably safe region. Additionally, we extend the applicability of our approach to incorporate the different gait parameters as contexts, leading to a safe, sample-efficient exploration algorithm capable of tuning a motion controller for diverse gait patterns. We validate our method through simulation and hardware experiments, where we demonstrate that the algorithm obtains superior performance on tuning a model-based motion controller for multiple gaits safely.

**Keywords:** Legged robot, Bayesian optimization, Safe learning, Controller tuning

## 1 Introduction

A model-based control strategy facilitates quick adaptation to various robots and eliminates the need for offline training, thereby streamlining the design and test phases. However, it requires an accurate dynamics model of the system, which is often unavailable due to our limited understanding of real-world physics and inevitable simplifications to reduce the computational burden. As a result, these controllers typically underperform on actual hardware without considerable parameter fine-tuning. This tuning process is not only time-consuming but can also harm the hardware platform. Additionally, it often requires reiteration for diverse environments or movement patterns.

This work explores the challenge of determining optimal control gain parameters for a model-based legged locomotion controller. In doing so, we aim to bridge the disparity between simplified models and actual hardware behavior, consequently improving the controller's robustness and tracking accuracy. To this end, we employ a *safe learning algorithm*, namely GOSAFEOPT [1] to automate the parameter tuning process, enabling the online identification of optimal control gain parameters within a safe region. Furthermore, we extend GOSAFEOPT by incorporating various gait parameters as *contexts* [2]. This facilitates more sample-efficient learning of control gains tailored for distinct gait patterns and allows for fluid online adjustments of the control gains during operation.

We demonstrate our method on the quadruped robot *Unitree Go1* [3] in both simulation and hardware experiments. In our simulation experiments, we show that contextual GOSAFEOPT outperforms other model-free safe exploration baselines while ensuring zero unsafe interactions. Moreover, when trained across varied gait patterns, the experimental results clearly indicate that our contextual GOSAFEOPT delivers a considerable performance boost. Moving to our hardware experiments, contextual GOSAFEOPT finds optimal feedback controller gains for both *trot* and *crawl* gaits in only

---

[*]These authors contributed equally.

7th Conference on Robot Learning (CoRL 2023), Atlanta, USA.

50 learning steps, *all while avoiding any unsafe interaction with the real robot*. The resulting controller gains, together with our model-based controller, ensure robust legged locomotion against perturbations and environmental uncertainties. In addition, our tests reveal that GOSAFEOPT can effectively suggest reasonably good controller gains for previously unseen gait patterns such as *flying trot* and *pronk*.

In summary, (*i*) we formulate the problem of safe control parameter tuning for a model-based legged locomotion controller as constrained optimization, (*ii*) we extend GOSAFEOPT to account for contextual scenarios while providing theoretical safety and optimality guarantees, (*iii*) we demonstrate the superiority of contextual GOSAFEOPT over other state-of-the-art safe exploration algorithms in supporting diverse gait patterns, and (*iv*) we show that our method successfully and safely tunes control gains on the hardware and enhances the robustness and tracking performance of the controller significantly.

## 2    Related Work

**Bridging the reality gap in legged locomotion tasks**    Several previous studies have emphasized the importance of considering an actuator behavior and identifying the system latency to successfully bridge the *reality gap* in legged robot systems [4, 5, 6]. These studies develop a simulation model of a legged robot system incorporating either modeled or learned actuator dynamics and train a control policy that can be effectively deployed to the robot hardware.

Incorporating this strategy into a model-based control framework is an area of active investigation. Rather, in the context of model-based control, it is typically more straightforward to introduce adjustable control gain parameters and fine-tune them to align with the real-world behaviors of the robot. For instance, Kim et al. [7] use joint position- and velocity-level feedback to joint torque command in order to address any discrepancy between the actual torque output and the intended torque command for robots with proprioceptive actuators [8]. However, the fine-tuning of these parameters continues to present a significant challenge. Schperberg et al. [9] utilize the unscented Kalman filter algorithm to recursively tune control parameters of a model-based motion controller online, and they successfully demonstrate it on the simulated quadrupedal robot in the presence of sensor noise and joint-level friction. However, their proposed tuning method is inherently unsafe and can therefore lead to arbitrary harmful interactions with the system. In contrast, our method aims to optimize control gains while avoiding any unsafe interactions with the robot hardware.

**Safe exploration for controller parameter tuning**    Training a controller directly on hardware is a challenging task, as it requires sample efficient and safe exploration to avoid possible damage to the robot. In such settings, Bayesian optimization (BO [10]) emerges as a suitable framework due to its sample efficiency. A notable example in the field of legged robotics comes from Calandra et al. [11], who successfully employed BO to learn optimal gait parameters for a bipedal robot platform.

BO methods can be easily adapted to constrained settings for safe learning. Gelbart et al. [12], Hernández-Lobato et al. [13], Marco et al. [14] utilize constrained BO for finding safe optimal controller parameters. However, these works do not provide safety assurance during exploration. In contrast, methods such as SAFEOPT [15, 16] and its extensions [1, 17, 18, 19] guarantee safety throughout the entire learning and exploration phases. SAFEOPT leverages regularity properties of the underlying optimization to expand the set of safe controllers. This expansion is inherently local, and accordingly SAFEOPT can miss the global optimum. For dynamical systems, Baumann et al. [19] introduced GOSAFE, a *global* safe exploration algorithm which, unlike SAFEOPT, is capable of identifying the global optimum. However, the BO routine proposed in GOSAFE is expensive and sample inefficient which limits the scalability of the method. To this end, Sukhija et al. [1] introduce GOSAFEOPT. GOSAFEOPT leverages the underlying Markovian structure of the dynamical system to overcome the GOSAFE's restrictions. As a result, it can perform global safe exploration for realistic and high-dimensional dynamical systems.

In this work, we extend GOSAFEOPT to a contextual setting and apply it to systematically tune the model-based controller of a quadruped robot for various gait patterns. Our proposed method not only guarantees safety and global optimality but also scales effectively to systems with relatively high-dimensional search space that involves a twenty-four-dimensional state space, six-dimensional parameter space, and five-dimensional context space.

# 3 Problem Setting

**Safe learning formulation**    The dynamics of robotic systems can generally be described as an ordinary differential equation (ODE) of the form $\dot{s} = f(s, u)$ where $u \in \mathcal{U} \subset \mathbb{R}^{d_u}$ is the control signal and $s \in \mathcal{S} \subset \mathbb{R}^{d_s}$ is the state of the robot. Due to the reality gap, disparities can arise between the real-world dynamics and the dynamics model $f$. This often results in a significant divergence between the behaviors of models and actual real-world systems, thereby making the control of intricate and highly dynamic systems like quadrupeds particularly challenging.

A common solution to this problem is using a feedback policy to rectify the model inaccuracies. Given a desired input signal $u^*$, desired state $s^*$, and true system state $s$, we formulate a parameterized feedback control policy in the form $u = \pi_\theta(u^*, s^*, s)$ that steers $s$ to closely align with $s^*$. The parameters $\theta$ are picked to minimize the tracking error. A common example of such a feedback policy is PD control, where $u = u^* + \theta(s^* - s)$, where $\theta \in \mathbb{R}^{d_u \times d_s}$ corresponds to the controller gains. Typically, choosing the parameters $\theta$ involves a heuristic process, requiring experimental iterations with the physical hardware. However, such interactions can be unpredictably risky and could possibly cause damage to the hardware.

In this work, we formalize the tuning process as a constrained optimization problem:

$$\max_{\theta \in \Theta} g(\theta) \quad \text{such that } q_i(\theta) \geq 0, \forall i \in \mathcal{I}_q, \tag{1}$$

where $g$ is an objective function (or reward function), $q_i$ are constraints with $\mathcal{I}_q = \{1, \ldots, c\}$, and $\Theta$ is a compact set of parameters over which we optimize. Since the true dynamics are unknown, we cannot solve Equation (1) directly. Instead, we interact with the robot to learn $g(\theta)$ and $q_i(\theta)$, and solve the optimization problem in a black-box fashion. As we interact directly with the robot hardware, it is important that the learning process is sample-efficient and safe, i.e., constraints $q_i$ are not violated during learning.

**Extension to a contextual setting**    Our goal is to find optimal control gains specific to individual gait patterns and facilitate seamless online transitions across various gaits. Each gait pattern demonstrates unique dynamic properties. Therefore, the optimal feedback parameters $\theta$ vary depending on the gait pattern in question. We consider gaits as *contexts* $z$ from a (not necessarily finite) set of contexts $\mathcal{Z}$ [2]. Contexts are essentially external variables specified by the user. We broaden our initial problem formulation from Equation (1) to accommodate these contexts;

$$\max_{\theta \in \Theta} g(\theta, z) \quad \text{such that } q_i(\theta, z) \geq 0, \forall i \in \mathcal{I}_q, \tag{2}$$

where $z \in \mathcal{Z}$ is the context, which in our scenario, is the parameters of the gait of interest.

**Assumptions**    We reiterate and discuss the assumptions for GOSAFEOPT [1] in Appendix A. To summarize, we assume the following: (*i*) an initial safe set of parameters is known, (*ii*) the objective and constraints lie in a reproducing kernel Hilbert space with bounded norm, (*iii*) measurement noises are i.i.d. sub-Gaussian, (*iv*) the control frequency is sufficiently high to capture the state evolution, and (*v*) constraints $q_i(\theta, z)$ can be defined as the minimum of a state-dependent function $\bar{q}_i(\theta, s, z)$ along the trajectory starting in $s_0$ with policy $\pi_\theta$.

# 4 Control Gain Optimization for Model-based Legged Locomotion Control

## 4.1 Control Pipeline

**Model-based locomotion controller**    Our locomotion controller utilizes a combination of the model predictive control (MPC) and the whole-body control (WBC) method following the previous work by Kim et al. [7], Kang et al. [20], and Kang et al. [21]. The MPC generates dynamically consistent base and foot trajectories by finding an optimal solution of a finite-horizon optimal control problem, using a simplified model. To convert these trajectories into joint-level control signals, we implement a WBC method that incorporates a more sophisticated dynamics model and takes into account the physical constraints of the robot. More specifically, we use a WBC formulation similar to the one presented by Kim et al. [7]. This method calculates the desired generalized coordinates $x^*$, speed $\dot{x}^*$, and acceleration

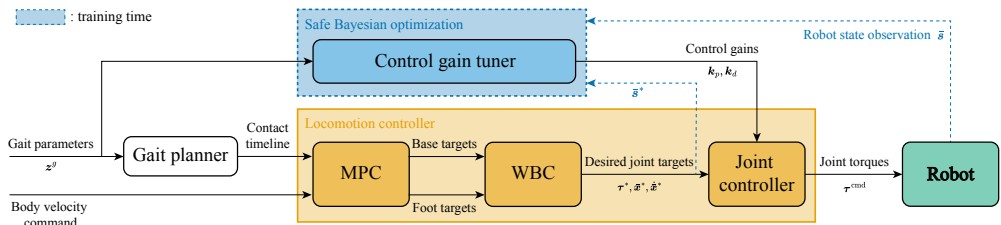

Figure 1: **Overview of the system.** The control gain tuner determines the optimal gains $\boldsymbol{k}_p, \boldsymbol{k}_d$ for the locomotion controller given gait parameters $\boldsymbol{z}^g$ as a context variable. In order to learn the map between the optimal gains and context variable, we use a safe Bayesian optimization algorithm, which finds optimal gains by minimizing the mismatch between desired joint states $\bar{\boldsymbol{s}}^*$ and actual joint states $\bar{\boldsymbol{s}}$ while ensuring no safety breach during the learning process.

$\ddot{\boldsymbol{x}}^*$ on a kinematic level while respecting task priority via the null-space projection [22]. Subsequently, it finds the desired joint torques $\boldsymbol{\tau}^*$ by solving a quadratic program that aligns with the desired generalized acceleration, adhering to the motion equations of the floating base and other physical constraints. For a more detailed explanation of the WBC formulation, the reader is referred to Appendix D.

We emphasize that the feed-forward torque commands $\boldsymbol{\tau}^*$ by themselves fail to produce the desired motion on the robot hardware due to model discrepancies. Particularly, we observed the actuator dynamics and joint friction, which are impractical to include in the system model, contribute significantly to this model mismatch. As a practical solution, we compute the final joint torque commands $\boldsymbol{\tau}^{\mathrm{cmd}} = \boldsymbol{\tau}^* + \boldsymbol{k}_p(\bar{\boldsymbol{x}}^* - \bar{\boldsymbol{x}}) + \boldsymbol{k}_d(\dot{\bar{\boldsymbol{x}}}^* - \dot{\bar{\boldsymbol{x}}})$ with the feedback gains $\boldsymbol{k}_p \in \mathbb{R}^{d_u \times d_{\bar{x}}}$ and $\boldsymbol{k}_d \in \mathbb{R}^{d_u \times d_{\bar{x}}}$ and send them to the robot. Here, $\bar{\boldsymbol{x}}, \dot{\bar{\boldsymbol{x}}}$ represent the joint angles and speeds (we use $\bar{\boldsymbol{s}}$ to represent the concatenated vector of $\bar{\boldsymbol{x}}$ and $\dot{\bar{\boldsymbol{x}}}$), while $\bar{\boldsymbol{x}}^*, \dot{\bar{\boldsymbol{x}}}^*$ denote their desired values. We treat the feedback gains $\boldsymbol{k}_p$ and $\boldsymbol{k}_d$ as the parameters $\boldsymbol{\theta}$ that we want to optimize using data samples collected from hardware directly.

**Gait parameterization** We parameterize a quadrupedal gait pattern with $\boldsymbol{z}^g = [d^g, t_s^g, o_1^g, o_2^g, o_3^g]$, where $d^g$ is the duty cycle for gait $g$, $t_s^g$ is the gait duration, and $o_i^g$ are the phase offsets of legs two to four respectively, starting counterclockwise with the rear left leg. The duty cycle is defined as the contact duration divided by the stride duration. In general, the optimal feedback parameters $(\boldsymbol{k}_p^*, \boldsymbol{k}_d^*)$ change with the gait. We show this empirically in Section 5.

## 4.2 Contextual GoSafeOpt

We model the unknown objective and constraint functions $h(\cdot, i)$ ($i = 0$ for the objective, $i \in \mathcal{I}_q$ for constraints) through Gaussian Process regression [23]. To this end, given a dataset $\{\boldsymbol{v}_j, \boldsymbol{y}_j\}_{j \leq n}$, with $\boldsymbol{v}_j = (\boldsymbol{\theta}_j, \boldsymbol{z}_j)$ and the kernel $k$, we calculate mean and uncertainty estimations of $h(\cdot, i)$:

$$\mu_n(\boldsymbol{v}, i) = \boldsymbol{k}_n^\top(\boldsymbol{v})(\boldsymbol{K}_n + \sigma^2 \boldsymbol{I})^{-1} \boldsymbol{y}_{n,i},$$
$$\sigma_n^2(\boldsymbol{v}, i) = k(\boldsymbol{v}, \boldsymbol{v}) - \boldsymbol{k}_n^\top(\boldsymbol{v})(\boldsymbol{K}_n + \sigma^2 \boldsymbol{I})^{-1} \boldsymbol{k}_n(\boldsymbol{v}),$$
(3)

where $\boldsymbol{y}_{n,i} = [y_{j,i}]_{j \leq n}^\top$ are the observations of $h(\cdot, i)$, $\boldsymbol{k}_n(\boldsymbol{v}) = [k(\boldsymbol{v}, \boldsymbol{v}_j)]_{j \leq n}^\top$, and $\boldsymbol{K}_n = [k(\boldsymbol{v}_j, \boldsymbol{v}_l)]_{j,l \leq n}$ is the kernel matrix. We leverage these estimates to provide high-probability frequentist confidence intervals.

**Lemma 1** (Confidence intervals, Theorem 2 of [24] and Lemma 4.1 of [16]). *Let $h$ be defined as*

$$h(\boldsymbol{\theta}, \boldsymbol{z}, i) = \begin{cases} g(\boldsymbol{\theta}, \boldsymbol{z}) & \text{if } i = 0, \\ q_i(\boldsymbol{\theta}, \boldsymbol{z}) & \text{if } i \in \mathcal{I}_q. \end{cases}$$
(4)

*For any $\delta \in (0, 1)$ and under Assumptions 2 and 3 from Appendix A, with probability at least $1 - \delta$ it holds jointly for all $n, i, \boldsymbol{z}, \boldsymbol{\theta}$ that*

$$|h(\boldsymbol{\theta}, \boldsymbol{z}, i) - \mu_n(\boldsymbol{\theta}, \boldsymbol{z}, i)| \leq \beta_n(\delta) \cdot \sigma_n(\boldsymbol{\theta}, \boldsymbol{z}, i)$$
(5)

*with $\beta_n(\delta) \leq \mathcal{O}(B + 4\sigma\sqrt{2(\gamma_{n|\mathcal{I}|} + 1 + \log(1/\delta))})$ where*

$$\gamma_n = \max_{\substack{A \subset \Theta \times \mathcal{Z} \times \mathcal{I} \\ |A| \leq n}} \mathrm{I}(\boldsymbol{y}_A; \boldsymbol{h}_A).$$
(6)

Here, $\mathrm{I}(\boldsymbol{y}_A; \boldsymbol{h}_A)$ denotes the *mutual information* between $\boldsymbol{h}_A = [h(\boldsymbol{v})]_{\boldsymbol{v} \in A}$, if modeled with a GP, and the noisy observations $\boldsymbol{y}_A$ at $\boldsymbol{h}_A$. It quantifies the reduction in uncertainty about $h$ upon observing $\boldsymbol{y}_A$ at points $A$. The quantity $\gamma_n$ is a Bayesian construct, however, in the frequentist setting it quantifies the complexity of learning the function $h$. It is instance-dependent and can be bounded depending on the domain $\Theta \times \mathcal{Z} \times \mathcal{I}$ and kernel function $k$ (see Appendix B).

Given the confidence interval from Equation (5), we define a confidence set for each context $\boldsymbol{z}$, parameter $\boldsymbol{\theta}$ and index $0 \leq i \leq c$, as

$$C_0(\boldsymbol{\theta}, \boldsymbol{z}, i) = \begin{cases} [0, \infty] & \text{if } \boldsymbol{\theta} \in \mathcal{S}_0(\boldsymbol{z}) \text{ and } i \geq 1, \\ [-\infty, \infty] & \text{otherwise,} \end{cases} \tag{7}$$

$$C_n(\boldsymbol{\theta}, \boldsymbol{z}, i) = C_{n-1}(\boldsymbol{\theta}, \boldsymbol{z}, i) \cap [\mu_n(\boldsymbol{\theta}, \boldsymbol{z}, i) \pm \beta_n(\delta) \cdot \sigma_n(\boldsymbol{\theta}, \boldsymbol{z}, i)], \tag{8}$$

We refer to $l_n(\boldsymbol{\theta}, \boldsymbol{z}, i) = \min C_n(\boldsymbol{\theta}, \boldsymbol{z}, i)$ as the lower bound, $u_n(\boldsymbol{\theta}, \boldsymbol{z}, i) = \max C_n(\boldsymbol{\theta}, \boldsymbol{z}, i)$ the upper bound, and $w_n(\boldsymbol{\theta}, \boldsymbol{z}, i) = u_n(\boldsymbol{\theta}, \boldsymbol{z}, i) - l_n(\boldsymbol{\theta}, \boldsymbol{z}, i)$ the width of our confidence set.

### 4.2.1 Algorithm

Given (user-specified) context $\boldsymbol{z}_n \in \mathcal{Z}$, an episode $n$ of contextual GOSAFEOPT is performed in one of two alternating stages: *local safe exploration* (LSE) and *global exploration* (GE).

**Local safe exploration** During the LSE stage, we explore the subset of the parameter space $\Theta$ which is known to be safe, and learn backup policies for all the states on the trajectories visited during LSE. In this stage, the parameters are selected according to the acquisition function

$$\boldsymbol{\theta}_n = \underset{\boldsymbol{\theta} \in \mathcal{G}_{n-1}(\boldsymbol{z}_n) \cup \mathcal{M}_{n-1}(\boldsymbol{z}_n)}{\operatorname{argmax}} \max_{i \in \mathcal{I}} w_{n-1}(\boldsymbol{\theta}, \boldsymbol{z}_n, i) \tag{9}$$

where, $\mathcal{G}_n(\boldsymbol{z}_n) \subseteq S_n(\boldsymbol{z}_n)$ is a set of *expanders* (c.f., Equation (16) in Appendix B) and $\mathcal{M}_n(\boldsymbol{z}_n) \subseteq S_n(\boldsymbol{z}_n)$ is a set of *maximizers* (c.f., Equation (18) in Appendix B). Intuitively, $\mathcal{G}_n(\boldsymbol{z}_n) \cup \mathcal{M}_n(\boldsymbol{z}_n)$ represents those parameters that can potentially lead to an expansion of the safe set $\mathcal{S}_n(\boldsymbol{z}_n)$ or potentially be a solution to the optimization problem of Equation (2) with context $\boldsymbol{z}_n$.

**Global exploration** Once LSE converges (see Equation (21) in Appendix B), we run the GE stage where we evaluate possibly unsafe policies and trigger a backup policy whenever necessary. If no backup policy is triggered, we conclude that the evaluated policy is safe and add it to our safe set. After a new parameter is added to the safe set during GE, we continue with LSE.

The parameters are selected according to the acquisition function

$$\boldsymbol{\theta}_n = \underset{\boldsymbol{\theta} \in \Theta \setminus (S_{n-1}(\boldsymbol{z}_n) \cup \mathcal{E}(\boldsymbol{z}_n))}{\operatorname{argmax}} \max_{i \in \mathcal{I}} w_{n-1}(\boldsymbol{\theta}, \boldsymbol{z}_n, i) \tag{10}$$

where $\mathcal{E}$ denotes all parameters which have been shown to be unsafe (see line 7 of Algorithm 4 in Appendix B). If all parameters have been determined as either safe or unsafe, i.e., $\Theta \setminus (S_n(\boldsymbol{z}_n) \cup \mathcal{E}(\boldsymbol{z}_n)) = \emptyset$, then GE has converged.

**Summary** A detailed description of the contextual GOSAFEOPT algorithm is provided in Appendix B.2. GOSAFEOPT alternates between local safe exploration and global exploration. Therefore, it can seek for the optimum globally. In Figure 5 of Appendix C, we analyze the algorithm using a simple example for better understanding.

The only difference between the contextual and non-contextual variants is that contextual GOSAFEOPT maintains separate sets $S_n, C_n, \mathcal{B}_n, \mathcal{D}_n, \mathcal{E}$, and $\mathcal{X}_{\mathrm{Fail}}$ for each context $\boldsymbol{z} \in \mathcal{Z}$. For any given context $\boldsymbol{z} \in \mathcal{Z}$, the running best guess of contextual GOSAFEOPT for the optimum is $\hat{\boldsymbol{\theta}}_n(\boldsymbol{z}) = \operatorname{argmax}_{\boldsymbol{\theta} \in S_n(\boldsymbol{z})} l_n(\boldsymbol{\theta}, \boldsymbol{z}, 0)$.

### 4.2.2 Theoretical Results

In the following, we state our main theorem, which extends the safety and optimality guarantees from Sukhija et al. [1] to the contextual case.

We say that the solution to Equation (2), $\boldsymbol{\theta}^*(\boldsymbol{z})$, is *discoverable* if there exists a finite $\tilde{n}$ such that $\boldsymbol{\theta}^*(\boldsymbol{z}) \in \bar{R}^{\boldsymbol{z}}_\epsilon(S_{\tilde{n}}(\boldsymbol{z}))$. Here, $\bar{R}^{\boldsymbol{z}}_\epsilon(S) \subseteq \Theta$ represents the largest safe set which can be reached safely from $S \subseteq \Theta$ up to $\epsilon$-precision (c.f., Equation (20) in Appendix B).

**Theorem 1.** *Consider any $\epsilon > 0$ and $\delta \in (0,1)$. Further, let Assumptions 1 to 5 from Appendix A hold and $\beta_n(\delta)$ be defined as in Lemma 1. For any context $\boldsymbol{z} \in \mathcal{Z}$, let $\tilde{n}(\boldsymbol{z})$ be the smallest integer such that*

$$\frac{n(\boldsymbol{z})}{\beta_{\tilde{n}(\boldsymbol{z})}(\delta) \cdot \gamma_{n(\boldsymbol{z})|\mathcal{I}|}(\boldsymbol{z})} \geq \frac{C|\Theta|^2}{\epsilon^2} \qquad where \quad n(\boldsymbol{z}) = \sum_{n=1}^{\tilde{n}(\boldsymbol{z})} \mathbb{1}\{\boldsymbol{z} = \boldsymbol{z}_n\} \tag{11}$$

*and $C = 32/\log(1 + \sigma^{-2})$. Here, $\gamma_n(\boldsymbol{z}) = \max_{A \subset \Theta \times \mathcal{I}, |A| \leq n} \mathrm{I}(\boldsymbol{y}_{A,\boldsymbol{z}}; \boldsymbol{h}_{A,\boldsymbol{z}}) \leq \gamma_n$ denotes the mutual information between $\boldsymbol{h}_{A,\boldsymbol{z}} = [h(\boldsymbol{\theta}, \boldsymbol{z}, i)]_{(\boldsymbol{\theta},i) \in A}$ and corresponding observations.*

*Then, when running contextual GOSAFEOPT and if $\boldsymbol{\theta}^*(\boldsymbol{z})$ is discoverable, the following inequalities jointly hold with probability at least $1 - 2\delta$:*

*1. $\forall n \geq 0, t \geq 0, i \in \mathcal{I}_q \colon \bar{q}_i(\boldsymbol{\theta}_n, \boldsymbol{s}(t), \boldsymbol{z}) \geq 0$,*                                                *(safety)*

*2. $\forall \boldsymbol{z} \in \mathcal{Z}, n \geq \tilde{n}(\boldsymbol{z}) \colon g(\hat{\boldsymbol{\theta}}_n(\boldsymbol{z}), \boldsymbol{z}) \geq g(\boldsymbol{\theta}^*(\boldsymbol{z}), \boldsymbol{z}) - \epsilon$.*                         *(optimality)*

It is natural to start for each $i \in \mathcal{I}$ with kernels $k_i^{\mathcal{Z}}$ and $k_i^\Theta$ on the space of contexts and the space of parameters, respectively, and to construct composite kernels $k_i = k_i^{\mathcal{Z}} \otimes k_i^\Theta$ or $k_i = k_i^{\mathcal{Z}} \oplus k_i^\Theta$ as the product or sum of the pairs of kernels (see section 5.1 of [2]). In this case, the information gain $\gamma_n$ is sublinear in $n$ for common choices of kernels $k_i^{\mathcal{Z}}$ and $k_i^\Theta$ implying that $n^*(\boldsymbol{z})$ is finite.

The theorem is proven in Appendix B.3. Comparing to contextual SAFEOPT [16] which is only guaranteed to converge to safe optima in $\bar{R}^{\boldsymbol{z}}_\epsilon(S_0(\boldsymbol{z}))$, the global exploration steps of contextual GOSAFEOPT can also *discover* a safe optimum which was not reachable from the initial safe seed. We remark that Theorem 1 is a worst-case result and, in particular, disregards a possible statistical dependence between different contexts. In practice, if a kernel is chosen which does not treat all contexts as independent, then the convergence can be much faster as knowledge about a particular context can be transferred to other contexts.

# 5 Experimental results

We evaluate the performance of contextual GOSAFEOPT using the *Unitree Go1* robot in both physical simulation and hardware experiments. In the experiments, we use the following objective and constraint functions:

$$g(\boldsymbol{\theta}, \boldsymbol{z}) = -\sum_{t \geq 0} \|\bar{\boldsymbol{s}}^*(t) - \bar{\boldsymbol{s}}(t, \boldsymbol{z}, \boldsymbol{\theta})\|^2_{\boldsymbol{Q}_g}, \quad q(\boldsymbol{\theta}, \boldsymbol{z}) = \min_{t \geq 0} v - \|\bar{\boldsymbol{s}}^*(t) - \bar{\boldsymbol{s}}(t, \boldsymbol{z}, \boldsymbol{\theta})\|^2_{\boldsymbol{Q}_q}, \tag{12}$$

where $\bar{\boldsymbol{s}}$ is joint-level states of the system (i.e. joint angles and speeds), $\bar{\boldsymbol{s}}^*$ denotes its desired values, and both $\boldsymbol{Q}_g$ and $\boldsymbol{Q}_q$ are positive semi-definite matrices. Additionally, we define an error threshold, $v$, that the norm of the state error shouldn't surpass throughout the entire duration. Further details of the experimental setup are provided in Appendix E. We also made the implementation available online[1] and uploaded a video showcasing the experiments[2].

**Simulation experiments** In our simulation experiments, we contrasted the learning curves of contextual GOSAFEOPT to SAFEOPT, and GOSAFEOPT without contexts. Additionally, we evaluate GP-UCB [25], an unconstrained BO algorithm. To simulate the model mismatches and uncertainties, we introduced disturbances in the form of joint impedances at every joint (see Appendix E for more details). These disturbances destabilize the system, leading to constraint violations. To adhere to the prerequisites of the safe BO algorithms, we initiate all experiments with roughly hand-tuned control gains that are safe, yet suboptimal.

---

[1] https://github.com/lasgroup/gosafeopt
[2] https://youtu.be/zDBouUgegrU

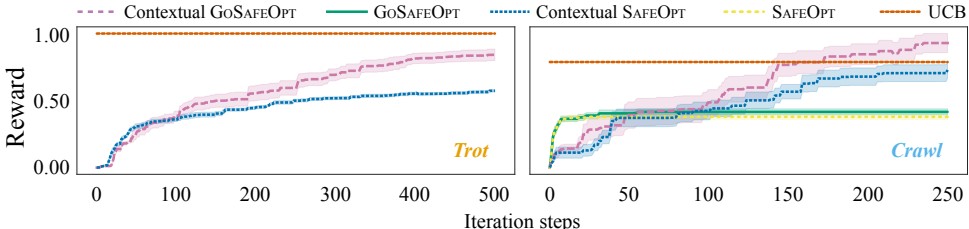

Figure 2: **Simulation experiments.** On the left, we display the learning curves of the BO algorithms trained with *trot* gait. After completing this training, we started new training for *crawl* gait and included the contextual variants of GOSAFEOPT and SAFEOPT in the assessment to investigate the impact of contextual settings on learning performance as illustrated on the right.

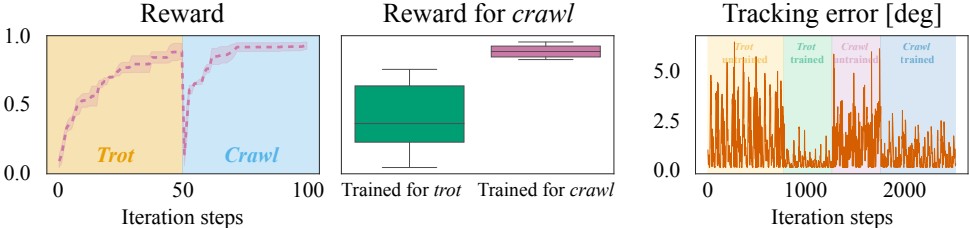

Figure 3: **Hardware experiments.** On the left, we present the learning curve of Contextual GOSAFEOPT. It shows that the algorithm successfully tunes the controller gains for *trot*, and then subsequently for *crawl*. In the center, we compare the performance of the optimized control gains of *trot* when applied to *crawl*, against the gains specifically optimized for *crawl*. On the right, we present the tracking error of the hip joint for the front-left leg with *trot* and *crawl* gait at initialization (*trot*: yellow, *crawl*: violet) and after optimization (*trot*: green, *crawl*: blue).

We optimized joint-level feedback gains for two different gaits; *trot* and *crawl* sequentially. All simulation experiments were conducted using ten different seeds, and we report the mean with one standard error in the Figure 2. Throughout our experiments, all of the safe algorithms met safety constraints. In contrast, the standard GP-UCB method violates the constraints in $4.7\%$ and $8\%$ of all evaluations for the *trot* and *crawl* gaits respectively. In Figure 2's left plot, we illustrate the normalized performance of GOSAFEOPT and SAFEOPT w.r.t. our objective for the *trot* gait. The learning curves clearly indicate that GOSAFEOPT's global exploration facilitates faster identification of better control gains. Notably, the GOSAFEOPT algorithm performs nearly as well as GP-UCB but without any constraint violations.

In the second test, we contrasted the contextual variants of GOSAFEOPT and SAFEOPT with their non-contextual counterparts. The results, as depicted in the right part of Figure 2, suggest that the contextual variants yield superior optima, with the contextual GOSAFEOPT algorithm emerging as the standout performer. The contextual variants leverage the information collected from the previous training with the *trot* gait, enabling them to identify better optima for the newly introduced *crawl* gait more efficiently. Additionally, they evade unsafe or unstable evaluations, unlike GP-UCB. The gait parameters we used for the *trot* and *crawl* gaits are provided in Appendix E.1.

**Hardware experiments**    Similarly to the simulation experiments, we first tune the controller for the *trot* gait and subsequently for the *crawl* gait. In Figure 3's left plot, we report the mean performance with one standard error, based on experiments conducted using three different seeds. In all our experiments, we note that the contextual GOSAFEOPT algorithm results in zero constraint violations.

In our hardware experiments, we confirmed that different gait patterns require distinct sets of control gains. As shown in the center plot of Figure 3, the best-performing parameters for the *trot* gait do not perform well on the *crawl* gait. However, as we train with a context for the new gait pattern *crawl*, there is a notable improvement in the reward. Here, we highlight that the contextual GOSAFEOPT can harness previously gathered data when encountering new gait patterns, accelerating the discovery of optimal gains for the new gait.

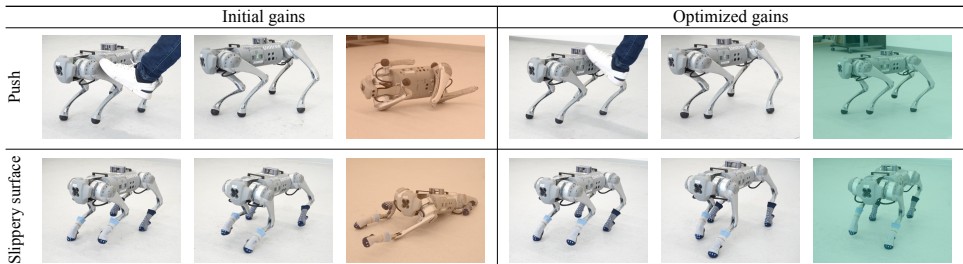

Figure 4: **Robustness test.** Compared to the roughly hand-tuned initial gains (left), the optimal gains derived from our method (right) significantly improve the motion controller's robustness against external pushes (top) and slippery contacts caused by socks on the robot's feet (bottom).

Additionally, in the right plot of Figure 3, we evaluate the tracking performance of our tuned controller, focusing on the hip joint's joint-angle tracking. When comparing the initial and the tuned controller across both *trot* and *crawl* gaits, it's evident that the tuned controller has a significantly reduced tracking error. For a comprehensive view, the error plots for other joints are provided in Appendix E.

To assess the robustness of the locomotion controller with the optimal control gains, we introduced uncertainties in the form of external forces and simulated slippery conditions by placing socks on the robot's feet, as depicted in Figure 4. Our experiments demonstrate that the robustness of our controller is significantly enhanced after the tuning process, and it is able to recover from pushes and retain stability on a slippery surface. On the other hand, the controller with the initial control gains is more vulnerable to these uncertainties and tends to easily crash.

Finally, we also highlight the zero-shot generalization capabilities of our method for unseen gait patterns through a learned model. While *flying trot* and *pronk* gaits were not presented during the training, the learned model effectively suggests reasonably good control gains for these gaits. We encourage readers to view the accompanying video[2] for a more in-depth understanding.

## 6    Conclusion

In this work, we extend GOSAFEOPT to the contextual setting and showcase its efficacy in adjusting the control parameters for a model-based legged locomotion controller through both simulation and hardware experiments. In our experiments, contextual GOSAFEOPT demonstrated superior convergence for newly introduced gait patterns by drawing upon information from previous training sessions. Additionally, our results confirmed that contextual GOSAFEOPT can effectively identify better optima without violating safety constraints. Across all of our experiments, contextual GOSAFEOPT outperforms prior approaches by a large margin and successfully finds optimal control gains for different quadrupedal gait patterns. After the fine-tuning process, we found that our model-based controller exhibits a considerable improvement in robustness to various types of uncertainties, significantly enhancing the system's reliability. We highlight that the applicability of our proposed algorithm extends beyond our current scenario. For instance, we are interested in applying this method to a more diverse set of quadrupedal gait patterns and extend its scope to encompass non-periodic and unstructured gait patterns. Furthermore, we stress that the algorithm is controller- or robot-agnostic, making it a pivotal tool for addressing the reality gap across various contexts.

**Limitations**    While the algorithm provides theoretical safety guarantees, it is often uncertain in real-world applications whether all theoretical prerequisites are fulfilled. For instance, even though the surrogate model might be Lipschitz-continuous, the Lipschitz constant is generally not known a priori, i.e., Assumption 2 from Appendix A may not be satisfied. This often results in a too conservative choice of parameters. Furthermore, a wrong parameter choice for the backup prior can result in unsafe global exploration or no global exploration at all. In general, while safe exploration methods such as SAFEOPT and GOSAFEOPT have been successfully applied on several practical domains [1, 17, 18, 26, 27, 28], bridging the disparity between theoretical foundations and real-world application remains a topic of active investigation [29, 30, 31].

## Acknowledgements

We thank Lenart Treven and Flavio De Vincenti for their feedback on this work.

This project has received funding from the Swiss National Science Foundation under NCCR Automation, grant agreement 51NF40 180545, the European Research Council (ERC) under the European Union's Horizon 2020 research and innovation programme, grant agreement No. 866480, and the Microsoft Swiss Joint Research Center.

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

# Contents of Appendix

# A  Assumptions

In this section, we reiterate the assumptions by Sukhija et al. [1] for GOSAFEOPT.

**Assumption 1** (Initial safe seed). *For any episode $n \geq 1$ with (user-specified) context $\boldsymbol{z}_n \in \mathcal{Z}$, a non-empty initial safe set of parameters $\mathcal{S}_{n-1}(\boldsymbol{z}_n) \subset \Theta$ is known. That is, for all $\boldsymbol{\theta} \in \mathcal{S}_{n-1}(\boldsymbol{z}_n)$ and all $i \in \mathcal{I}_q$, $q_i(\boldsymbol{\theta}, \boldsymbol{z}_n) \geq 0$.*

Here, $\mathcal{S}_n(\boldsymbol{z}) \supseteq \mathcal{S}_0(\boldsymbol{z})$ denotes the safe set after episode $n$ for the given context $\boldsymbol{z}$ as defined in Equation (15) in Appendix B. Given the prior knowledge of the dynamics, a conservative safe set of parameters represents some initial stable feedback controller. Accordingly, this assumption is typically satisfied in practice. The assumption is necessary as, in principle, during each iteration, an adversarial context could be chosen for which the initial safe set does not include any safe parameters.

**Assumption 2** (Continuity of objective and constraints). *Let $h$ be defined as*

$$h(\boldsymbol{\theta}, \boldsymbol{z}, i) = \begin{cases} g(\boldsymbol{\theta}, \boldsymbol{z}) & \text{if } i = 0, \\ q_i(\boldsymbol{\theta}, \boldsymbol{z}) & \text{if } i \in \mathcal{I}_q. \end{cases} \tag{13}$$

*We assume that $h$ lies in a reproducing kernel Hilbert space (RKHS) associated with a kernel $k$ and has a bounded norm in that RKHS, that is, $\|h\|_k \leq B$. Furthermore, we assume that $g$ and $q_i$ ($\forall i \in \mathcal{I}_q$) are Lipschitz-continuous with known Lipschitz constants.*

This is a common assumption in the model-free safe exploration literature [16, 19, 1]. Sukhija et al. [1] discuss the practical implications of this assumption in more detail.

**Assumption 3.** *We obtain noisy measurements of $h$ with measurement noise i.i.d. $\sigma$-sub-Gaussian. Specifically, for a measurement $y_i$ of $h(\boldsymbol{\theta}, \boldsymbol{z}, i)$, we have $y_i = h(\boldsymbol{\theta}, \boldsymbol{z}, i) + \epsilon_i$ with $\epsilon_i$ $\sigma$-sub-Gaussian for all $i \in \mathcal{I}$ where we write $\mathcal{I} = \{0, \ldots, c\}$.*

**Assumption 4.** *We observe the state $\boldsymbol{s}(t)$ every $\Delta t$ seconds. Furthermore, for any $\boldsymbol{s}(t)$ and $\rho \in [0, 1]$, the distance to $\boldsymbol{s}(t + \rho \Delta t)$ induced by any action is bounded by a known constant $\Xi$, that is, $\|\boldsymbol{s}(t + \rho \Delta t) - \boldsymbol{s}(t)\| \leq \Xi$.*

Assumption 4 is crucial to guarantee safety in continuous time even though the state is measured at discrete time instances. For highly dynamical systems, such as quadrupeds, the observation frequency is typically very high, e.g., $500\,\text{Hz}$ - $1\,\text{kHz}$, and accordingly $\Xi$ is small.

**Assumption 5.** *We assume that, for all $i \in \{1, \ldots, c\}$, $q_i$ is defined as the minimum of a state-dependent function $\bar{q}_i$ along the trajectory starting in $\boldsymbol{s}_0$ with controller $\boldsymbol{\pi}_{\boldsymbol{\theta}}$. Formally,*

$$q_i(\boldsymbol{\theta}, \boldsymbol{z}) = \min_{\boldsymbol{s}' \in \xi_{(\boldsymbol{s}_0, \boldsymbol{\theta}, \boldsymbol{z})}} \bar{q}_i(\boldsymbol{s}', \boldsymbol{z}, \boldsymbol{\theta}), \tag{14}$$

*with $\xi_{(\boldsymbol{s}_0, \boldsymbol{\theta}, \boldsymbol{z})} = \{\boldsymbol{s}_0 + \int_0^t \boldsymbol{f}(\boldsymbol{s}(\tau), \boldsymbol{\pi}_{\boldsymbol{\theta}}(\boldsymbol{s}(\tau), \boldsymbol{z}))\, d\tau \mid t \geq 0\}$ representing the trajectory of $\boldsymbol{s}(t)$ under policy parameter $\boldsymbol{\theta}$ and context $\boldsymbol{z}$ starting from $\boldsymbol{s}_0$ at time $0$.*

Assumption 5 is an assumption on our choice of the constraint. Many common constraints, such as the minimum distance to an obstacle along a trajectory, satisfy this assumption.

# B  Proofs

## B.1  Definitions

We begin by re-stating definitions of sets used by GOSAFEOPT [1] with an additional context variable.

Fix an arbitrary context $\boldsymbol{z} \in \mathcal{Z}$. The *safe set* is defined recursively as

$$S_n(\boldsymbol{z}) = \bigcap_{i \in \mathcal{I}_q} \bigcup_{\boldsymbol{\theta}' \in S_{n-1}(\boldsymbol{z})} \{\boldsymbol{\theta} \in \Theta \mid l_n(\boldsymbol{\theta}', \boldsymbol{z}, i) - L_\Theta(\boldsymbol{z}) \|\boldsymbol{\theta} - \boldsymbol{\theta}'\| \geq 0\} \tag{15}$$

where $L_\Theta(z)$ is the joint Lipschitz constant of $g$ and the constraints $q_i$ under context $z$. The *expanders* are defined as

$$\mathcal{G}_n(z) = \{\boldsymbol{\theta} \in S_n(z) \mid e_n(\boldsymbol{\theta}, z) > 0\} \qquad \text{with} \tag{16}$$

$$e_n(\boldsymbol{\theta}, z) = |\{\boldsymbol{\theta}' \in \Theta \setminus S_n(z) \mid \exists i \in \mathcal{I}_q \colon u_n(\boldsymbol{\theta}, z, i) - L_\Theta(z) \|\boldsymbol{\theta} - \boldsymbol{\theta}'\| \geq 0\}| \tag{17}$$

and the *maximizers* are defined as

$$\mathcal{M}_n(z) = \{\boldsymbol{\theta} \in S_n(z) \mid u_n(\boldsymbol{\theta}, 0) \geq \max_{\boldsymbol{\theta}' \in S_n(z)} l_n(\boldsymbol{\theta}', 0)\}. \tag{18}$$

The analysis requires the $\epsilon$-slacked safe region $\bar{R}_\epsilon^z(S)$ given an initial safe seed $S \subseteq \Theta$, which is defined recursively as

$$R_\epsilon^z(S) = S \cup \{\boldsymbol{\theta} \in \Theta \mid \exists \boldsymbol{\theta}' \in S \text{ such that } \forall i \in \mathcal{I}_q \colon q_i(\boldsymbol{\theta}', z) - \epsilon - L_\Theta(z) \|\boldsymbol{\theta} - \boldsymbol{\theta}'\| \geq 0\}, \tag{19}$$

$$\bar{R}_\epsilon^z(S) = \lim_{n \to \infty} (R_\epsilon^z)^n(S) \tag{20}$$

where $(R_\epsilon^z)^n$ denotes the $n$th composition of $R_\epsilon^z$ with itself.

## B.2 Algorithm

### B.2.1 Local Safe Exploration

During LSE, we keep track of a set of backup policies $\mathcal{B}(z) \subseteq \Theta \times \mathcal{X}$ and observations of $h$ for each context $z \in \mathcal{Z}$, which we denote by $\mathcal{D}(z) \subseteq \Theta \times \mathbb{R}^{|\mathcal{I}|}$. An LSE step is described formally in Algorithm 1.

---

**Algorithm 1** Local Safe Exploration (LSE)

---

**Input**: Current context $z_n$, safe sets $S$, sets of backups $\mathcal{B}$, datasets $\mathcal{D}$, Lipschitz constants $L_\Theta$
 1: Recommend parameter $\boldsymbol{\theta}_n$ with Equation (9)
 2: Collect $\mathcal{R} = \bigcup_{k \in \mathbb{N}} \{(\boldsymbol{\theta}_n, \boldsymbol{s}(k))\}$ and $h(\boldsymbol{\theta}_n, z_n, i) + \varepsilon_n$
 3: $\mathcal{B}(z_n) = \mathcal{B}(z_n) \cup \mathcal{R}$, $\mathcal{D}(z_n) = \mathcal{D}(z_n) \cup \{(\boldsymbol{\theta}_n, h(\boldsymbol{\theta}_n, z_n, i) + \varepsilon_n)\}$
 4: Update sets $S(z)$, $\mathcal{G}(z)$, and $\mathcal{M}(z)$ for all $z \in \mathcal{Z}$    ▷ Equations (15), (16) and (18)
**Return**: $S, \mathcal{B}, \mathcal{D}$

---

The LSE stage terminates for some given context $z \in \mathcal{Z}$ when the connected safe set is fully explored and the optimum within the safe set is discovered. This happens when the uncertainty among the expanders and maximizers is less than $\epsilon$ and the safe set is not expanding

$$\max_{\boldsymbol{\theta} \in \mathcal{G}_{n-1}(z) \cup \mathcal{M}_{n-1}(z)} \max_{i \in \mathcal{I}} w_{n-1}(\boldsymbol{\theta}, z, i) < \epsilon \qquad \text{and} \qquad S_{n-1}(z) = S_n(z). \tag{21}$$

### B.2.2 Global Exploration

A GE step conducts an experiment about a candidate parameter $\boldsymbol{\theta}_n \in \Theta$ which may not be safe. If the safety boundary is approached, GE conservatively triggers a safe backup policy. If, on the other hand, the experiment is successful, a new (potentially disconnected) safe region was discovered which can then be explored by LSE in the following steps. A GE step is described formally in Algorithm 2.

### B.2.3 Boundary Condition

The boundary condition checks when the system is in the state $\boldsymbol{s}$ whether there is a backup $(\boldsymbol{\theta}_s, \boldsymbol{s}_s) \in \mathcal{B}(z)$ such that $\boldsymbol{s}_s$ is sufficiently close to $\boldsymbol{s}$ to guarantee that $\boldsymbol{\theta}_s$ can steer the system back to safety for any state which may be reached in the next time step. If no such backups exist for the next states, a backup is triggered at the current state. In this case, the backup parameter $\boldsymbol{\theta}_s^*$ with the largest safety margin is triggered:

$$\boldsymbol{\theta}_s^* = \max_{(\boldsymbol{\theta}_s, \boldsymbol{s}_s) \in \mathcal{B}_n(z_n)} \min_{i \in \mathcal{I}_q} l_n(\boldsymbol{\theta}_s, z_n, i) - L_x \|\boldsymbol{s} - \boldsymbol{s}_s\|. \tag{22}$$

---

**Algorithm 2** Global Exploration (GE)

---

**Input**: $\boldsymbol{z}_n$, safe sets $S$, confidence intervals $C$, sets of backups $\mathcal{B}$, datasets $\mathcal{D}$, fail sets $\mathcal{E}$ and $\mathcal{X}_{\text{Fail}}$

1: Recommend global parameter $\boldsymbol{\theta}_n$ with Equation (10)
2: $\boldsymbol{\theta} = \boldsymbol{\theta}_n$, $\boldsymbol{s}_{\text{Fail}} = \emptyset$, Boundary = False
3: **while** Experiment not finished **do**                                           ▷ Rollout policy
4:    **if** Not Boundary **then**
5:        Boundary, $\boldsymbol{\theta}_s^* = \text{BOUNDARYCONDITION}(\boldsymbol{z}_n, \boldsymbol{s}(k), \mathcal{B})$
6:        **if** Boundary **then**                                              ▷ Trigger backup policy
7:            $\boldsymbol{\theta} = \boldsymbol{\theta}_s^*$, $\boldsymbol{s}_{\text{Fail}} = \boldsymbol{s}(k)$
8:            $\mathcal{E} = \mathcal{E} \cup \{\boldsymbol{\theta}_n\}$, $\mathcal{X}_{\text{Fail}} = \mathcal{X}_{\text{Fail}} \cup \{\boldsymbol{s}_{\text{Fail}}\}$                 ▷ Update fail sets
9:    Execute until $\boldsymbol{s}(k)$
10: Collect $\mathcal{R} = \bigcup\limits_{k \in \mathbb{N}} \{(\boldsymbol{\theta}_n, \boldsymbol{s}(k))\}$, and $h(\boldsymbol{\theta}_n, \boldsymbol{z}_n, i) + \varepsilon_n$
11: **if** Not Boundary **then**                                        ▷ Successful global search
12:    $\mathcal{B}(\boldsymbol{z}_n) = \mathcal{B}(\boldsymbol{z}_n) \cup \mathcal{R}$ and $\mathcal{D}(\boldsymbol{z}_n) = \mathcal{D}(\boldsymbol{z}_n) \cup \{(\boldsymbol{\theta}_n, h(\boldsymbol{\theta}_n, \boldsymbol{z}_n, i) + \varepsilon_n)\}$
13:    $S(\boldsymbol{z}_n) = S(\boldsymbol{z}_n) \cup \{\boldsymbol{\theta}_n\}$
14:    $C(\boldsymbol{\theta}_n, \boldsymbol{z}_n, i) = C(\boldsymbol{\theta}_n, \boldsymbol{z}_n, i) \cap [0, \infty]$ for all $i \in \mathcal{I}_q$

**Return**: $S, C, \mathcal{B}, \mathcal{D}, \mathcal{E}, \mathcal{X}_{\text{Fail}}$

---

---

**Algorithm 3** BOUNDARYCONDITION

---

**Input**: context $\boldsymbol{z}_n$, state $\boldsymbol{s}$, backups $\mathcal{B}$

1: **if** $\forall (\boldsymbol{\theta}_s, \boldsymbol{s}_s) \in \mathcal{B}(\boldsymbol{z}_n), \exists i \in \mathcal{I}_q : l_n(\boldsymbol{\theta}_s, \boldsymbol{z}_n, i) - L_x \|\boldsymbol{s} - \boldsymbol{s}_s\| + \Xi < 0$ **then**
2:    Boundary = True, Calculate $\boldsymbol{\theta}_s^*$ (Equation (22))
3: **else**
4:    Boundary = False, $\boldsymbol{\theta}_s^* = \text{Null}$

**return**: Boundary, $\boldsymbol{\theta}_s^*$

---

### B.2.4   Contextual GOSAFEOPT

The algorithm stops for a particular context $\boldsymbol{z} \in \mathcal{Z}$ when

$$\underbrace{\text{Equation (21) is satisfied}}_{\text{LSE converged}} \quad \text{and} \quad \underbrace{\Theta \setminus (S_n(\boldsymbol{z}_n) \cup \mathcal{E}(\boldsymbol{z}_n)) = \emptyset}_{\text{GE converged}}. \tag{23}$$

The full algorithm is described in Algorithm 4.

---

**Algorithm 4** Contextual GOSAFEOPT

---

**Input**: Domain $\Theta$, Contexts $\mathcal{Z}$, Sequence of contexts $\{\boldsymbol{z}_n \in \mathcal{Z}\}_{n \geq 1}$, $k(\cdot, \cdot)$, $S_0, C_0, \mathcal{D}_0, \epsilon$

1: Initialize GP $h(\boldsymbol{\theta}, \boldsymbol{z}, i)$, $\mathcal{E}(\boldsymbol{z}) = \emptyset$, $\mathcal{X}_{\text{Fail}}(\boldsymbol{z}) = \emptyset$, $\mathcal{B}_0(\boldsymbol{z}) = \{(\boldsymbol{\theta}, x_0) \mid \boldsymbol{\theta} \in S_0\}$
2: **while** $\exists \boldsymbol{z} \in \mathcal{Z}$ such that GOSAFEOPT has not terminated for $\boldsymbol{z}$ (Equation (23)) **do**
3:    **if** GOSAFEOPT has terminated for $\boldsymbol{z}_n$ (Equation (23)) **then**          ▷ Skip finished contexts
4:        **continue**
5:    **for** $\boldsymbol{s} \in \mathcal{X}_{\text{Fail}}(\boldsymbol{z}_n)$ **do**                                     ▷ Update fail sets
6:        **if** Not BOUNDARYCONDITION$(\boldsymbol{z}_n, \boldsymbol{s}, \mathcal{B}_n)$ **then**
7:            $\mathcal{E}(\boldsymbol{z}_n) = \mathcal{E}(\boldsymbol{z}_n) \setminus \{\boldsymbol{\theta}\}$, $\mathcal{X}_{\text{Fail}}(\boldsymbol{z}_n) = \mathcal{X}_{\text{Fail}}(\boldsymbol{z}_n) \setminus \{\boldsymbol{s}\}$
8:    Update $C_n(\boldsymbol{\theta}, \boldsymbol{z}, i) \forall \boldsymbol{\theta} \in \Theta, \boldsymbol{z} \in \mathcal{Z}, i \in \mathcal{I}$      ▷ Update confidence intervals, Equation (8)
9:    **if** LSE not converged for context $\boldsymbol{z}_n$ (Equation (21)) **then**
10:        $S_{n+1}, \mathcal{B}_{n+1}, \mathcal{D}_{n+1} = \text{LSE}(\boldsymbol{z}_n, \mathcal{S}_n, \mathcal{B}_n, \mathcal{D}_n)$
11:    **else**
12:        $S_{n+1}, C_{n+1}, \mathcal{B}_{n+1}, \mathcal{D}_{n+1}, \mathcal{E}, \mathcal{X}_{\text{Fail}} = \text{GE}(\boldsymbol{z}_n, \mathcal{S}_n, C_n, \mathcal{B}_n, \mathcal{D}_n, \mathcal{E}, \mathcal{X}_{\text{Fail}})$

**return**: $\{\hat{\boldsymbol{\theta}}_n(\boldsymbol{z}) \mid \boldsymbol{z} \in \mathcal{Z}\}$

---

## B.3 Proof of Theorem 1

*Proof.* We first derive the sample complexity bound of non-contextual GOSAFEOPT. Then, we extend this sample complexity bound to contextual GOSAFEOPT. We assume without loss of generality that $\beta_n$ is monotonically increasing with $n$.

**Sample complexity**  Assume first that the context is fixed, that is, $\forall n \geq 1 : z_n = z$. In this case, the safety guarantee (with probability at least $1 - \delta$) follows directly from Theorem 4.1 of Sukhija et al. [1]. Thus, it remains to show that the optimality guarantee with the given sample complexity holds also with probability at least $1 - \delta$, as then their union holds jointly with probability at least $1 - 2\delta$ using a union bound.

It is straightforward to see (by employing Theorem 4.1 of Berkenkamp et al. [16]) that Theorem 4.2 of Sukhija et al. [1] holds for $n^*$ being the smallest integer such that

$$n^* \geq \frac{C|\Theta|\beta_{n^*}(\delta)\gamma_{n^*|\mathcal{I}|}}{2\epsilon^2} \tag{24}$$

where we use that $|\bar{R}_0^z(S)| \leq |\Theta|$ for any $S \subseteq \Theta$ and $|\Theta| + 1 \leq 2|\Theta|$. Thus, whenever a new disconnected safe region is discovered by GE, LSE is run for at most $n^*$ steps.

It follows from the stopping criterion of GE, $\Theta \setminus (S_n \cup \mathcal{E}) = \emptyset$, that GE is run for at most $|\Theta|$ consecutive steps (i.e., without an LSE-step in between). Clearly, a new disconnected safe region can be discovered by GE at most $|\Theta|$ times, and hence, GOSAFEOPT terminates after at most $|\Theta|$ iterations of at most $n^*$ LSE steps and at most $|\Theta|$ GE steps. Altogether, we have that the optimality guarantee holds with probability at least $1 - \delta$ for $\tilde{n}$ being the smallest integer such that

$$\tilde{n} = \left\lceil \frac{C|\Theta|^2\beta_{\tilde{n}}(\delta)\gamma_{\tilde{n}|\mathcal{I}|}}{\epsilon^2} \right\rceil \geq |\Theta|\,(n^* + |\Theta|), \tag{25}$$

completing the proof of Theorem 1 for non-contextual GOSAFEOPT.

**Multiple contexts**  Visiting other contexts $\mathcal{Z} \setminus \{z\}$ in between results in additional measurements and increases the constant $\beta$, ensuring that the confidence intervals are well-calibrated. The only difference in the proofs is the appearance of $\beta_{n^*(z)}$ rather than $\beta_{n(z)}$ in Equation (24). In the contextual setting, $n^*(z)$ is the smallest integer such that

$$n(z) \geq \frac{C|\Theta|\beta_{n^*(z)}(\delta)\gamma_{n(z)|\mathcal{I}|}(z)}{2\epsilon^2} \tag{26}$$

where

$$n(z) = \sum_{n=1}^{n^*(z)} \mathbb{1}\{z = z_n\}$$

counts the number of episodes with context $z$ until episode $n^*(z)$. The bound on $\tilde{n}(z)$ then follows analogously to Equation (25). $\qquad\square$

## C  Comparison of SAFEOPT and GOSAFEOPT

To visually analyze the different exploration properties of SAFEOPT and GOSAFEOPT we use the Pendulum Environment from OpenAI [33] as an example. The ideal trajectory is given by some undisturbed controller. In our toy problem, we use a simple PD control which is sufficient for the pendulum swing-up problem and various oscillating trajectories. To simulate the sim to hardware gap, we artificially add a disturbance to the applied torque in the form of joint impedances $\boldsymbol{\tau} = \boldsymbol{\tau}^* - \boldsymbol{\theta}_p^d(\tilde{\boldsymbol{x}}^* - \tilde{\boldsymbol{x}}) + \boldsymbol{\theta}_d^d\dot{\tilde{\boldsymbol{x}}}$ where $\boldsymbol{\theta}_p^d$ and $\boldsymbol{\theta}_d^d$ are unknown disturbance parameters and $\tilde{\boldsymbol{x}}^*, \tilde{\boldsymbol{x}}$ are the desired and observed motor angles. We use GOSAFEOPT to tune an additional PD controller which should follow the ideal trajectory and compensate for the artificial disturbance. Figure 5 shows an example run of SAFEOPT and GOSAFEOPT. Whereas SAFEOPT is restricted to expanding the initial safe region, GOSAFEOPT can discover new safe regions, and thus find a better optimum.

Table 1: Here, we summarize different magnitudes of $\gamma_n$ for composite kernels from Theorems 2 and 3 of Krause and Ong [2] and for individual kernels from Theorem 5 of Srinivas et al. [25] and Remark 2 of Vakili et al. [32]. The magnitudes hold under the assumption that the domain of the kernel is compact. $\gamma^\Theta$ and $\gamma^\mathcal{Z}$ denote the information gain for the kernels $k^\Theta$ and $k^\mathcal{Z}$, respectively. $B_\nu$ is the modified Bessel function.

| Kernel | $k(\boldsymbol{v}, \boldsymbol{v}')$ | $\gamma_n$ |
|---|---|---|
| Product | $k^\Theta(\boldsymbol{v}, \boldsymbol{v}') \cdot k^\mathcal{Z}(\boldsymbol{v}, \boldsymbol{v}')$ if $k^\mathcal{Z}$ has rank at most $d$ | $d\gamma_n^\Theta + d\log(n)$ |
| Sum | $k^\Theta(\boldsymbol{v}, \boldsymbol{v}') + k^\mathcal{Z}(\boldsymbol{v}, \boldsymbol{v}')$ | $\gamma_n^\Theta + \gamma_n^\mathcal{Z} + 2\log(n)$ |
| Linear | $\boldsymbol{v}^\top \boldsymbol{v}'$ | $\mathcal{O}\left(d\log(n)\right)$ |
| RBF | $e^{-\frac{\|\boldsymbol{v}-\boldsymbol{v}'\|^2}{2l^2}}$ | $\mathcal{O}\left(\log^{d+1}(n)\right)$ |
| Matérn | $\frac{1}{\Gamma(\nu)2^{\nu-1}}\left(\frac{\sqrt{2\nu}\|\boldsymbol{v}-\boldsymbol{v}'\|}{l}\right)^\nu B_\nu\left(\frac{\sqrt{2\nu}\|\boldsymbol{v}-\boldsymbol{v}'\|}{l}\right)$ | $\mathcal{O}\left(n^{\frac{d}{2\nu+d}}\log^{\frac{2\nu}{2\nu+d}}(n)\right)$ |

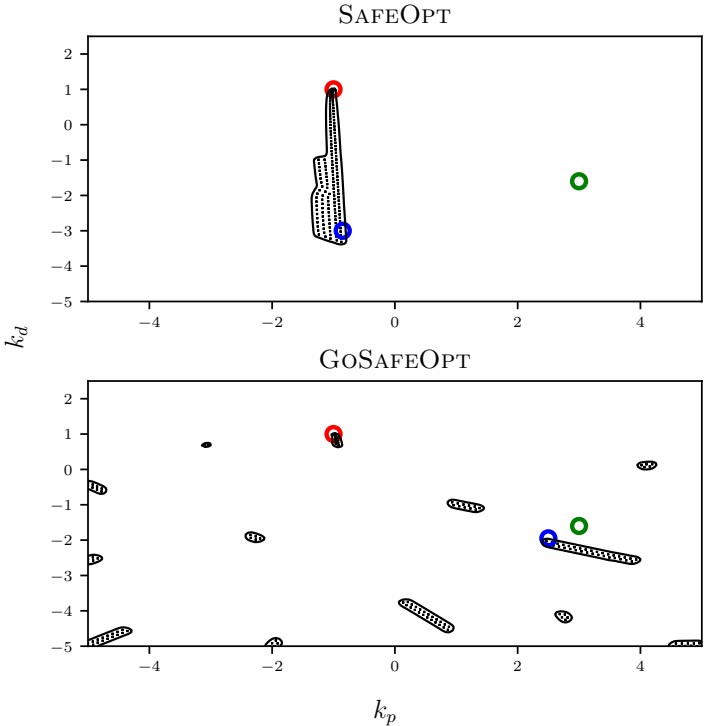

Figure 5: **Example run of SAFEOPT and GOSAFEOPT.** The red circle denotes the initial safe point. The black dots denote observed points. The green circle denotes the true safe optimum and the blue circle denotes the optimal point determined by SAFEOPT and GOSAFEOPT after 150 iterations respectively. The discovered safe sets are shown in black. GOSAFEOPT gets closer to the true optimum by discovering new safe regions which are not connected to the initial safe region.

## D  Control Formulation

Our model-based motion controller integrates the MPC and WBC methods to enhance both robustness and maneuverability. The MPC is responsible for generating base and foot trajectories, while the WBC converts these trajectories into joint-level commands. For the MPC component, we employ the model predictive control formulation proposed by Kang et al. [21, 34]. This formulation represents a

finite-horizon optimal control problem as a nonlinear program utilizing the variable-height inverted pendulum model. The optimal solution of the nonlinear program is determined by using a second-order gradient-based method. For a more in-depth understanding of the MPC formulation, we direct readers to the prior work by Kang et al. [21, 34].

We employ a slight modification of the WBC formulation introduced by Kim et al. [7], adapting it to align with our MPC method. Following the method proposed by Kim et al. [7], we compute the desired generalized coordinates $\boldsymbol{x}^*$, speed $\dot{\boldsymbol{x}}^*$, and acceleration $\ddot{\boldsymbol{x}}^*$ for a quadruped system at the kinematic level. This process involves translating desired task space (Cartesian space) positions, velocities, and accelerations into configuration space counterparts. Throughout this process, we enforce task priority through iterative null-space projection [22]. The top priority is assigned to the contact foot constraint task, followed by the base orientation tracking task. The base position tracking task is given the third priority, and the swing foot tracking task is assigned the final priority.

Subsequently, we solve the following quadratic program:

$$\min_{\boldsymbol{\delta}_{\ddot{x}}, \boldsymbol{f}_c} \quad \|\boldsymbol{\delta}_{\ddot{x}}\|_{\boldsymbol{Q}}^2 \tag{27a}$$

$$\text{s.t.} \quad \boldsymbol{S}_f(\boldsymbol{M}\ddot{\boldsymbol{x}} + \boldsymbol{b} + \boldsymbol{g}) = \boldsymbol{S}_f \boldsymbol{J}_c^\top \boldsymbol{f}_c \tag{27b}$$

$$\ddot{\boldsymbol{x}}^{**} = \ddot{\boldsymbol{x}}^* + \left[\boldsymbol{\delta}_{\ddot{x}}, \boldsymbol{0}_{n_j}\right]^\top \tag{27c}$$

$$\boldsymbol{W}\boldsymbol{f}_c \geq \boldsymbol{0}, \tag{27d}$$

where $\boldsymbol{\delta}_{\ddot{x}}$ denotes a relaxation variables for the floating base acceleration and $\boldsymbol{f}_c$ denotes contact forces with the contact Jacobian $\boldsymbol{J}_c$. Equation (27a) is the objective function that penalizes the weighted norm of $\boldsymbol{\delta}_{\ddot{x}}$ with the weight matrix $\boldsymbol{Q}$. Equation (27b) corresponds to the equation of motion of the floating base, representing the first six rows of the whole-body equation of motion, with $\boldsymbol{S}_f$ being the corresponding selection matrix. Lastly, Equation (27d) sets forth the Coulomb friction constraints. This procedure refines the desired generalized acceleration $\ddot{\boldsymbol{x}}^*$, which is calculated at the kinematic level, by incorporating the dynamic impacts of the robot's movements.

Upon determining $\ddot{x}$ is determined, we compute the joint torque commands as follows:

$$\boldsymbol{\tau}^* = \boldsymbol{M}\ddot{\boldsymbol{x}}^{**} + \boldsymbol{b} + \boldsymbol{g} - \boldsymbol{J}_c^\top \boldsymbol{f}_c. \tag{28}$$

The final torque commands are calculated using $\boldsymbol{\tau}^{\text{cmd}} = \boldsymbol{\tau}^* + \boldsymbol{k}_p(\bar{\boldsymbol{x}}^* - \bar{\boldsymbol{x}}) + \boldsymbol{k}_d(\dot{\bar{\boldsymbol{x}}}^* - \dot{\bar{\boldsymbol{x}}})$ and dispatched to the robot with the feedback gains $\boldsymbol{k}_p \in \mathbb{R}^{d_u \times d_{\bar{x}}}$ and $\boldsymbol{k}_d \in \mathbb{R}^{d_u \times d_{\bar{x}}}$. Here, $\bar{\boldsymbol{x}}, \dot{\bar{\boldsymbol{x}}}$ are the joint angles and speeds, while $\bar{\boldsymbol{x}}^*, \dot{\bar{\boldsymbol{x}}}^*$ denote their desired values. As previously noted, this step is crucial in dealing with model mismatches, specifically, the differences in joint-level behavior stemming from actuator dynamics and joint friction.

## E  Experimental Details

### E.1  Gait parameters

We used the following gait parameters for the simulation and the hardware experiments.

Table 2: Gait parameters.

|  | Trot | Crawl | Flying trot | Pronk |
|---|---|---|---|---|
| Duration [s] | 0.5 | 1.2 | 0.6 | 0.6 |
| Duty cycle | 0.5 | 0.75 | 0.4 | 0.9 |
| Phase Offsets | 0.5 | 0.25 | 0.5 | 0.0 |
|  | 0.5 | 0.5 | 0.5 | 0.0 |
|  | 0 | 0.75 | 0 | 0.0 |

## E.2 Bayesian optimization

For all our experiments, we use a Matérn kernel with $\nu = 1.5$ for the underlying Gaussian Process. The lengthscales are fixed during the whole optimization process and set to

Table 3: Kernel lengthscales.

|  | lengthscales |
|---|---|
| Simulation | $[0.1, 0.05, 0.1, 0.05, 0.1, 0.05, 0.1, 0.05, 0.1, 0.1, 0.1, 0.1, 0.1]$ |
| Hardware | $[0.3, 0.3, 0.3, 0.3, 0.3, 0.3, 0.5, 0.5, 0.5, 0.5, 0.5]$ |

where the first $n$ parameters correspond to the $(\boldsymbol{k}_p, \boldsymbol{k}_d)$ pairs and the last parameters to the context. For all experiments, we use $\beta = 16$ for the LCB on the constraints.

## E.3 Simulation

We developed an emulator to simulate the control gain tuning process, utilizing the open-source rigid-body simulation engine, the Open Dynamics Engine (ODE) [35]. To account for model mismatches and uncertainties, we introduced disturbances based on the model detailed in subsequent sections.

**Disturbance model**  We introduced joint-level disturbances by altering the torque exerted by each motor. This method emulates the torque tracking discrepancies in motors, the damping effects stemming from joint friction, and other model mismatches attributed to inaccuracies in the model. More specifically, for $i$th motor of leg $l$, the applied motor torque is given by $\boldsymbol{\tau}_{i,l}^{\text{applied}} = \alpha_l \boldsymbol{\tau}_{i,l}^{\text{cmd}} + \boldsymbol{\theta}_l^\top [\bar{\boldsymbol{x}}_{l,i}^* - \bar{\boldsymbol{x}}_{l,i}, -\dot{\bar{\boldsymbol{x}}}_{l,i}]^\top$. In this equation, $\boldsymbol{\tau}^{\text{cmd}}$ is the torque command computed as described in Appendix D, $\alpha_l$ is a disturbance factor for leg $l$ with $\alpha = [0.73, 0.9, 0.73, 0.9]^\top$, and $\bar{\boldsymbol{x}}, \dot{\bar{\boldsymbol{x}}}$ are the joint angles and speeds, while $\bar{\boldsymbol{x}}^*, \dot{\bar{\boldsymbol{x}}}^*$ denote their desired values.

**Reward function**  The joint states variable $\bar{\boldsymbol{s}} \in \mathbb{R}^{24}$ of all 12 joints is described as a concatenated vector of joint angles and joint speeds. We set the matrices from Equation (12) to $\boldsymbol{Q}_g = \boldsymbol{I}^{24 \times 24}$ and $\boldsymbol{Q}_q^{i,j} = \mathbb{1}\{i = j \wedge i <= 12\}$.

## E.4 Hardware

We slightly modify the reward function for the hardware experiment and include a penalty term on the joint velocities.

$$\hat{g}(\boldsymbol{\theta}, \boldsymbol{z}) = g(\boldsymbol{\theta}, \boldsymbol{z}) - \|\bar{\boldsymbol{s}}(t)\|_{\boldsymbol{Q}_p}^2,$$

where the velocity state errors in $\boldsymbol{Q}_g$ and $\boldsymbol{Q}_q$ in Equation 12 are set to zero, since the joint speed measurements are noisy finite difference approximations of the joint angles. Furthermore, we define $\boldsymbol{Q}_p$ to only include the noisy joint speeds observations. More specifically, we define $\boldsymbol{Q}_q^{i,j} = \boldsymbol{Q}_q^{i,j} = \mathbb{1}\{i = j \wedge i <= 12\}$ and $\boldsymbol{Q}_p^{i,j} = \frac{1}{2}\mathbb{1}\{i = j \wedge i > 12\}$ and $\boldsymbol{Q}_q, \boldsymbol{Q}_q, \boldsymbol{Q}_p \in \mathbb{R}^{24 \times 24}$. Experimental results have shown, that adding a penalty term on the joint velocities acts as a regulator to prefer solutions where motor vibrations are low. This has shown to improve overall convergence and to visibly avoid solutions where motor vibrations are high.

Figure 6 shows that the optimal feedback control parameters drastically reduce motor vibrations and increase the tracking performance.

# F  Practical modifications

## F.1  Boundary conditions

We use the idea from Sukhija et al. [1] to reduce computational complexity by defining an interior and marginal set. Intuitively, the interior set contains all observed states for which the safety margin is

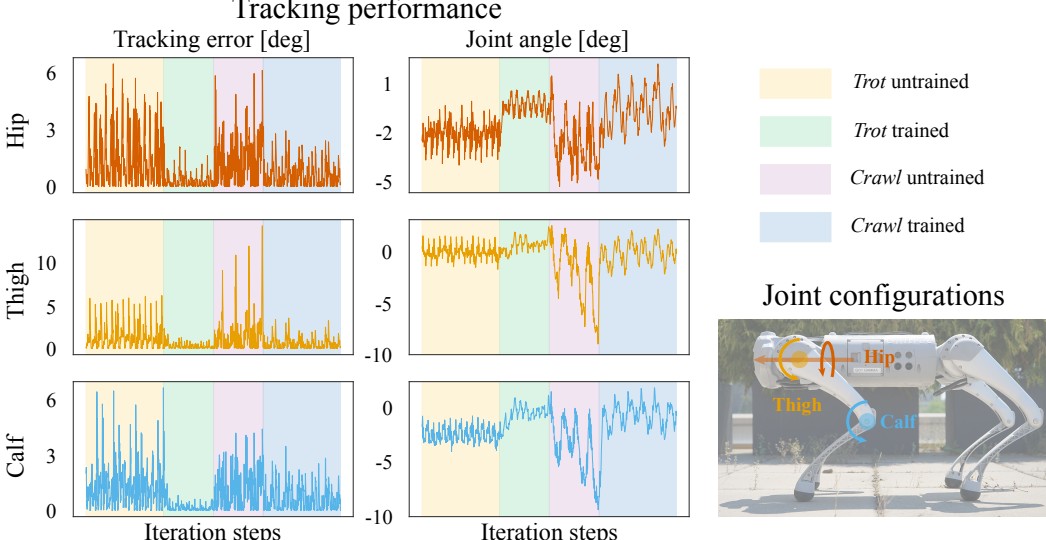

Figure 6: **Joint angle tracking performance comparison.** Joint angle tracking errors in degrees (left) and joint angle measurements in degrees (right.) The yellow and violet regions represent the initial control gains for *trot* and *crawl* respectively. Conversely, the green and blue regions indicate the optimized gains for *trot* and *crawl*. It is evident from the plots that the refined gains yield a substantially reduced tracking error with diminished jitter.

high and the marginal set includes all states where the safety margin is greater than a certain threshold. More formally, Sukhija et al. [1] defines the interior and marginal set as :

$$\Omega_{I,n} = \{\boldsymbol{x}_s \in \mathcal{X} \mid (\boldsymbol{\theta}, \boldsymbol{x}_s) \in \mathcal{B}_n : \forall i \in \mathcal{I}_q, l_n(\boldsymbol{\theta}, i) \geq \eta_u\} \tag{29}$$

$$\Omega_{M,n} = \{\boldsymbol{x}_s \in \mathcal{X} \mid (\boldsymbol{\theta}, \boldsymbol{x}_s) \in \mathcal{B}_n : \forall i \in \mathcal{I}_q, \eta_l \leq l_n(\boldsymbol{\theta}, i) < \eta_u\} \tag{30}$$

The boundary condition is defined separately for the interior and marginal set. Firstly, the Euclidean distance $d_i$ between the observed state and all the backup states is calculated. If $d_{\min} = \min_i d_i = 0$, a backup policy for the observed state is known to be safe. Intuitively, the uncertainty if a backup policy can safely recover from the observed state increases as $d_{\min}$ grows. If the observed state moves too far away from the set of backup states, the closest backup policy is triggered. More formally, a backup policy is triggered, if the $\nexists d_i$ s.t $p(|x| \geq d_i) > \tau$. The distribution over $x$ is defined as $x \sim \mathcal{N}(0, \sigma^2)$ and $\tau_m \geq \tau_i$ for the interior and marginal set, respectively. With $\sigma^2$ and $\tau_i$ there are two adjustable parameters to influence how conservative the backup policy acts.

Table 4: Boundary condition parameters

| Parameter | Value | Description |
|---|---|---|
| Simulation | | |
| $\sigma$ | 2 | Standard deviation of backup distribution |
| $\tau_i$ | 0.2 | Interior lower bound probability |
| $\tau_m$ | 0.6 | Marginal lower bound probability |
| Hardware | | |
| $\sigma$ | 2 | Standard deviation of backup distribution |
| $\tau_i$ | 0.05 | Interior lower bound probability |
| $\tau_m$ | 0.1 | Marginal lower bound probability |

### F.2 Optimization

The solution of the acquisition optimization problem formulated in 10 is approximated with the standard particle swarm [36] algorithm, similar to [37].

At the beginning of each acquisition optimization, $n_p$ particle positions are initialized. Rather than initializing the positions over the whole domain, the positions are sampled from a list of known safe positions in the current safe set.

For all experiments, the parameters in Table 5 are used.

Table 5: Swarmopt parameters

| Parameter | Value | Description |
|---|---|---|
| $\Theta_g$ | 1 | Social coefficient |
| $\Theta_p$ | 1 | Cognitive coefficient |
| $w$ | 0.9 | Inertial weight |
| $n$ | 100 | Number of iterations |
| $n_r$ | 100 | Number of restarts if no safe set is found |

### F.3 Fix iterations and discard unpromising new detected safe regions

In practice, it is not practical to fully explore a safe set before the global exploration phase. For our experiments, the number of iterations for the local and global exploration phase are fixed to $n_l = 10$ and $n_g = 5$, respectively. To avoid exploring for all $n_l$ steps in unpromising regions, we define $n_d = 5 < n_l$ and switch to local exploration of the best set if the best reward estimation of the current set is much less than the best global reward estimate. That is, we switch to the best set if $\hat{r}_i^* < c\hat{r}^*$ and $n_d = n_l$.

### F.4 Posterior estimation

Each BO step requires the optimization of the GOSAFEOPT acquisition function to predict the next parameters to evaluate. This paper uses the standard particle swarm [36] algorithm, which requires the computation of the posterior distribution at each optimization step for all particles. To speed up the computation of the posterior distribution, the paper uses Lanczos Variance Estimates Pleiss et al. [38].

