# OpenReview forum: "Tuning Legged Locomotion Controllers via Safe Bayesian Optimization"
_robot-learning.org/CoRL/2023/Conference — CoRL 2023 Poster_

### Official Review · Reviewer_2qzt · 2023-07-13

**Confidence:** 3
**Originality:** Very Good
**Technical Quality:** Very Good
**Clarity Of Presentation:** Very Good
**Impact:** 3

**Recommendation:**

Weak Accept: I recommend accepting the paper, but will not argue for my recommendation if the majority of other reviewers have a different opinion.

**Review:**

The paper is presented with both theoretical results and hardware results and addresses the important question of how to fine-tune controllers safely on real robots, so I believe this is a good contribution to the robot learning community.

My only concern is that the controllers presented are already working pretty well before tuning, and the algorithm seems to only need to do minimal tuning to further improve them. It will be interesting to see more interesting situations where the original controller can barely work.

**Quality Of The Limitations Section:**

Limitations are addressed clearly

**Questions For Rebuttal:**

1. It will be interesting to see how the gains evolve during the optimization process to understand how much work the optimization need to do.

2. In the demo video, there is a clip (starting at around 22 seconds) where a controller fails to directly transfer to the hardware, however, this is not used as a starting point for fine-tuning. Can the authors elaborate on if it is possible to fine-tune from that controller and if possible present relevant results?

**Robotics Focus:**

Sufficient demonstration on hardware

**Summary Of Paper:**

This paper presents an algorithm to learn optimal feedback control gains for model-based quadruped controllers. Theoretical results are presented to demonstrate the proposed method's optimality and safety, and hardware experiments are used to support the claims.

**Summary Of Recommendation:**

This paper presents an algorithm to safely tune control parameters on real robots, with sufficient theoretical and hardware results. I believe this can be valuable to the robot learning community and recommend acceptance.

---

### Official Review · Reviewer_bsr3 · 2023-07-17

**Confidence:** 4
**Originality:** Good
**Technical Quality:** Good
**Clarity Of Presentation:** Very Good
**Impact:** 3

**Recommendation:**

Weak Accept: I recommend accepting the paper, but will not argue for my recommendation if the majority of other reviewers have a different opinion.

**Review:**

**Strengths:**

1. Convincing demonstration on real hardware

In the supplemental video, the authors clearly demonstrated the effectiveness of the proposed method. Tuning the feedback gains (as well as other parameters) in optimal-control-based controllers is a long and tedious process, and I truly appreciate the authors' effort in developing an automated and safe approach to this problem.

2. Clarity of presentation

Overall, I find the paper to be clearly written and easy to follow. I especially appreciate the fact that the authors added explanations and intuitions next to the assumptions and theorems, which makes the paper very easy to read and understand. The experimental results is also concise and convincing.

**Weaknesses:**

1. Incremental contribution on algorithms

If I understand correctly, the proposed method is an extension of GoSafeOpt with contextual information (in this case gaits). While the authors performed comprehensive evaluation of the proposed method, I find the overall algorithm contribution to be relatively incremental and straightforward, because the inclusion of context in Bayesian Optimization (e.g. Contextual Bandits or [CGP-UCB](https://papers.nips.cc/paper_files/paper/2011/hash/f3f1b7fc5a8779a9e618e1f23a7b7860-Abstract.html)) has been proposed in previous studies.

2. More diverse context information would be desirable

In this paper, the "context" seems to be mostly restrained to the gait of the robot, while in practice many other factors might affect the controllers' performance, such as speed command and terrain type. Would it be beneficial to include these information in the context vector? How to encode these information in the context vector? Would the proposed algorithm still work well with the expanded context?

**Quality Of The Limitations Section:**

Limitations are addressed clearly

**Questions For Rebuttal:**

Please refer to the weakness section in my review.

**Robotics Focus:**

Sufficient demonstration on hardware

**Summary Of Paper:**

The paper presents a method to safely and automatically tune parameters (feedback gains) for legged locomotion controllers. The underlying algorithm is an extension of GoSafeOpt[1] that includes contextual information (gait parameters). The authors provide both theoretical and experimental validation for the proposed method, and shows that parameters tuned using GoSafeOpt can increase the overall stability and smoothness of the resulting controller.

**Summary Of Recommendation:**

Overall I find this to be a clear, well-written paper with well-demonstrated theoretical and experimental results. Therefore, I recommend acceptance for this paper. I'm holding back from a strong accept because I find the theoretical contribution to be relatively incremental and the experimental validation to be more of a proof-of-concept and does not fully capture the complexity of real-world.

---

### Official Review · Reviewer_9Nt1 · 2023-07-20

**Confidence:** 4
**Originality:** Good
**Technical Quality:** Fair
**Clarity Of Presentation:** Very Good
**Impact:** 2

**Recommendation:**

Weak Accept: I recommend accepting the paper, but will not argue for my recommendation if the majority of other reviewers have a different opinion.

**Review:**

The paper takes a previous method, extends it with contexts, shows the safety guarantees and applies it to quadruped problem. The intro, motivation and the overall idea is very interesting. As the authors state, the sim-to-real gap cause by the model discrepancies makes MPC hard to use for some problems.

The authors do a great job at introducing the problem and the method. But the author's contribution is very incremental and mainly composed of having context. The idea of applying to quadruped locomotion is interesting, because MPC is known to work quite well for these quadrupeds as the model is well studied.

The formulation of the problem is very similar to the PD controller and setting the gains. So I'm a little confused by the usage of MPC here which outputs desired torques. If I understand correctly, the authors convert it to position, then they track the position while tuning the gains?

The idea of safe exploration is very interesting and the results show significant improvement. On the other hand, the problems the author study is limited to 2 different gaits for walking. I think the method had great potential to overcome different behaviors such as hopping, jumping or very fast galloping behaviors.

I think that the paper has a high potential, but it can benefit from a problem where the model is unknown or very hard to model (i.e. attach a bucket filled with water?) and different behaviors where safety is critical while MPC needs to push the boundaries of the robot. In its current state, the results show behaviors that can be already accomplished.

**Quality Of The Limitations Section:**

Limitations are addressed clearly

**Questions For Rebuttal:**

Can the authors clarify the above question on torque vs pd controls? Does the controller purely a position control?

Can the method handle unkown latency in the system?

**Robotics Focus:**

Sufficient demonstration on hardware

**Summary Of Paper:**

The paper proposes a safe an efficient method to tune model of a robot based on the hardware data. This allows running MPC on a more accurate model. The authors extend a previous algorithm to have context and apply it to quadruped walking in different gaits. The authors describe the algorithm and its safety guarantees in details. The proposed method is tested on quadrupeds both in sim and in real.

**Summary Of Recommendation:**

I recommend weak reject because of the evaluation. The claims and theory is well explained, the contribution is incremental, but the evaluation lacks the domain and experiments to prove the claims further. Given further experiments or domains the paper can be very significant, but it is not well supported in its current form.

---

### Official Review · Reviewer_2pRp · 2023-07-21

**Confidence:** 4
**Originality:** Good
**Technical Quality:** Fair
**Clarity Of Presentation:** Good
**Impact:** 3

**Recommendation:**

Weak Reject: I recommend rejecting the paper, but will not argue for my recommendation if the majority of other reviewers have a different opinion.

**Review:**

While the proposed method is interesting, it is a highly complex approach to implement a rather basic feature: tuning PD gains for a well-known torque control loop. A lot of assumptions are made (e.g. noise in Section 3.1) to ensure safe exploration. However, the experimental section does not validate these assumptions.

The method is evaluated on a very simple test case (basic trotting and crawling on flat terrain) rather than with state-of-the-art policies.
The manuscript mentions that "Even though we have a good model of the robot dynamics, the motors are typically difficult to model sufficiently accurately." This is true. However, there are multiple recent methods available to address this issue (e.g. actuator net), which could be used as a comparison to ground the results.

Finally, the method appears incremental (adding a context variable to GoSafeOpt) and beyond the basic experiments in the paper, the value of adding the context variable is not clear.

**Quality Of The Limitations Section:**

Additional details required

**Questions For Rebuttal:**

- Hardware results: It's not clear how the data was collected to optimize the gains? Was this in sim and then deployed to real or did you iteratively improve the performance on hardware?
- Consider demonstrating the usefulness of the method with state of the art skills (jumping, agile skills) and in non-ideal scenarios (disturbances, uneven terrain etc.).
- How long are the rollouts? The paper mentions iteration steps throughout: Are these timesteps or complete rollouts?
- When does the method fail?
- The hardware parameter space was rather small (6D). How does the method scale to a higher number of dimensions?
- Please compare to existing methods (RMA, Actuator Net, manually tuning gains etc.) to make it possible to understand the relative performance of the proposed method.

**Robotics Focus:**

Sufficient demonstration on hardware

**Summary Of Paper:**

This paper introduces a data-driven method to safely and automatically tune controller feedback gains.
The authors validate their work by deploying an MPC style policy on an Go1 robot and tuning the torque control loop's gains.
On the theoretical side, the work builds on a prior method called GoSafeOpt and extends it by introducing a notion of context, making it possible to discover gait-specific gains.


**Summary Of Recommendation:**

The method in itself is interesting, however it is highly complex to achieve a relatively basic outcome: tuning PD gains.
The main hardware result is that by tuning the PD gains of the torque control loop in a gait-specific way, the joint position tracking error is lower and the robot is less noisy as a result of less over/undershoot during deployment.
However, the evaluation is done on very basic gaits under perfect conditions. Go1 is capable of highly agile skills, yet this work focuses on basic trotting and crawling behaviors, which make it hard to assess its value with respect to the current state of the art.
The theoretical contribution appears incremental (adding a context variable to GoSafeOpt for each gait).

---

### Comment · Area_Chair_m8GF · 2023-08-10
**Response to reviewers**

Dear Authors,

please submit your responses to reviewers on Openreview soon to enable a constructive discussion with reviewers during the rebuttal time window.

Best,

AC

---

> ### Author Response · Authors · 2023-08-10
> **Response to area chair**
>
> Dear area chair,
>
> We are sorry for the delay. Based on the reviewers’ feedback we performed some additional experiments to demonstrate the performance of our approach. With this, we believe that we have addressed the main concerns of our reviewers and also showcased the strengths of our method. We are looking forward to further fruitful exchanges with our reviewers.
>
> Thank you.

---

### Decision · Program_Chairs · 2023-08-30

**Decision:**

Accept (Poster)

**Comment:**

The paper presents a method for safe and automatic parameter tuning of feedback gains in a legged locomotion controller. The underlying algorithm is an extension of GoSafeOpt that includes contextual information. The authors provide both theoretical and experimental validation for the proposed method. Please prepare your camera-ready paper by including the results of additional experiments you made during the rebuttal phase and the clarifications discussed with reviewers.